# Magnetic resonance colonography assessment of acute trinitrobenzene sulfonic acid colitis in pre-pubertal rats

Claire Dupont-Lucas[1¤]*, Rachel Marion-Letellier[1], Mathilde Pala[1], Charlène Guerin[1], Christine Bôle-Feysot[1], Emmeline Salameh[1], Alexis Goichon[1], Lionel Nicol[2], Moutaz Aziz[3], Céline Savoye-Collet[4], Guillaume Savoye[5]

1 INSERM UMR 1073, Institute for Biomedical Research, Rouen University, Rouen, France, 2 INSERM UMR 1096, Institute for Biomedical Research, Rouen University, Rouen, France, 3 Department of Pathology, Rouen University Hospital, Rouen, France, 4 QUANTIF-LITIS EA 4108, Rouen University, Rouen, France, 5 Department of Gastroenterology, Rouen University Hospital, Rouen, France

¤ Current address: Department of Pediatrics, Caen University Hospital, Caen, France
* dupont-c@chu-caen.fr

**Data Availability Statement:** The dataset is available as supplementary material: https://figshare.com/s/01b2dd941d08b0fa6d9e).

## Abstract

Pre-pubertal murine models of acute colitis are lacking. Magnetic resonance colonography (MRC) is a promising minimally invasive tool to assess colitis. We aimed to: 1/ Adapt a model of acute experimental colitis to pre-pubertal rats and determine whether MRC characteristics correlate with histological inflammation. 2/ Test this model by administering a diet supplemented in transforming growth factor β2 to reverse inflammation. Twenty-four rats were randomized at weaning to one of 3 groups: Trinitrobenzene Sulfonic Acid (TNBS) group (n = 8) fed a standard diet, that received an intra-rectal 60 mg/kg dose of TNBS-ethanol; Control group (n = 8) fed standard diet, that received a dose of intra-rectal PBS; TNBS+MODULEN group (n = 8) that received a dose of TNBS and were exclusively fed MODULEN-IBD® after induction of colitis. One week after induction of colitis, rats were assessed by MRC, colon histopathology and inflammation markers (Interleukin 1β, Tumor necrosis factor α, Nitric Oxide Synthase 2 and Cyclooxygenase 2). TNBS induced typical features of acute colitis on histopathology and MRC (increased colon wall thickness, increased colon intensity on T2-weighted images, target sign, ulcers). Treatment with MODULEN-IBD® did not reduce signs of colitis on MRC. Inflammatory marker expression did not differ among study groups.

## Introduction

Inflammatory bowel disease (IBD), including Crohn's disease and Ulcerative colitis, is a chronic relapsing disease affecting the digestive tract. The incidence and prevalence of these diseases are increasing worldwide [1]. A systematic worldwide review showed that the highest reported prevalence values for IBD were in Europe (UC: 505 per 100,000 persons, CD: 322 per 100,000 persons) [1]. An estimated ten percent of new IBD cases patients are children, with a steady increase of incidence of pediatric IBD worldwide [2, 3]. Crohn's disease beginning in childhood has several specificities compared to adult-onset disease among which a higher incidence of complicated phenotypes and of growth failure [4–6]. Focusing on the mechanisms of

**Funding:** This work was supported by the European Union and Normandy Regional Council. Europe contributes to Normandy through the European Regional Development Fund (ERDF). One of the authors (MP) received a grant from Nestlé Health Science given by SFNCM (Société Francophone Nutrition Clinique et Métabolisme). The funders had no role in study design, data collection and analysis, decision to publish, or preparation of the manuscript.

**Competing interests:** No authors have competing interests.

the similarities and differences between pediatric and adult IBD could help unravel some aspects of IBD pathogenesis.

Although the cause of IBD remains unknown, studies provide evidence that pathogenesis of disease involves a loss of immune tolerance to the gut microbiota in a genetically susceptible host, exposed to environmental factors [7]. Animal models of intestinal inflammation mimicking Inflammatory Bowel Disease (IBD) are widely used to better understand the cellular and molecular pathways of inflammation and fibrosis, and target these pathways to develop new drugs [8–11]. Among these, the 2,4,6 Trinitrobenzene Sulfonic Acid Colitis (TNBS) model is frequently used and many variations exist [12]. Intra-rectal instillation of TNBS causes an IL-12-driven Th1 T-cell mediated immune response in the colonic mucosa [13]. Briefly, TNBS is instilled intra rectally mixed with ethanol, which acts as a mucosal barrier breaker allowing the hapten TNBS to interact with colonic proteins and elicit an immunological response by rendering these proteins immunogenic to the host immune system. The stimulated antigen producing cells secrete IL-12, causing induction of IFN-γ by the T cells, which stimulates macrophages to produce inflammatory mediators such as TNF-α, IL-6 and IL-1β. The resulting colonic inflammation resembles Crohn's disease, with transmural inflammation, ulcers and granulomas. The clinical picture of TNBS colitis is also similar to Crohn's disease, associating weight loss and bloody diarrhea [13]. The model has been used to test anti-inflammatory properties of dietary compounds, and drugs on colonic inflammation through inhibition of the NF-κB pathway [11, 14].

Most studies on acute TNBS colitis in rats have been carried out on adult animals. However weanling and pre-pubertal rats have several distinct features that would require adapting the TNBS model, such as rapid growth and susceptibility to growth failure, increased mucosal permeability and increased susceptibility to drugs [15, 16].

In order to follow non-invasively the effect of interventions and reduce number of animals needed for longitudinal studies, small animal imaging techniques have been developed. In a model of acute TNBS colitis in adult rats we previously showed that magnetic resonance colonography (MRC) could accurately evaluate inflammation, compared to histopathology [17].

The recommended treatment for inducing remission in children with IBD is exclusive enteral nutrition [18]. Several enteral diets have been compared, among which a polymeric diet enriched in transforming growth factor beta 2 (TGF-beta 2): MODULEN-IBD® (Nestle, Vevey, Switzerland). Administered as sole food source for 8 weeks, MODULEN-IBD® can induce clinical remission in 79% children and mucosal healing in 31% [19].

The objectives of our study were: 1/ to adapt a model of acute experimental colitis to prepubertal rats and determine whether MRC characteristics could be correlated to histopathology. 2/ To test this model by treating the rats with MODULEN-IBD® in the aim of reversing inflammation.

## Materials and methods

### Study design

Twenty-four Sprague Dawley male rats were purchased at weaning (postnatal day 21) from Janvier labs (Le Genest Saint Isle, France). Mean baseline weight was 77.4 ± 3.9 g. They were randomly allocated to one of three study groups: Control (n = 8), TNBS (n = 8) and TNBS + MODULEN (n = 8) (Fig 1, Panel A). The rats were housed 4 per standard cage to provide for their interaction needs, were exposed to light / dark cycles of 12 hours each and provided with water *ad libitum*. After induction of colitis, the TNBS + MODULEN group received MODULEN-IBD® powder as sole food source. The other groups received a standard rat breeding diet (A03, SAFE) in powdered form.

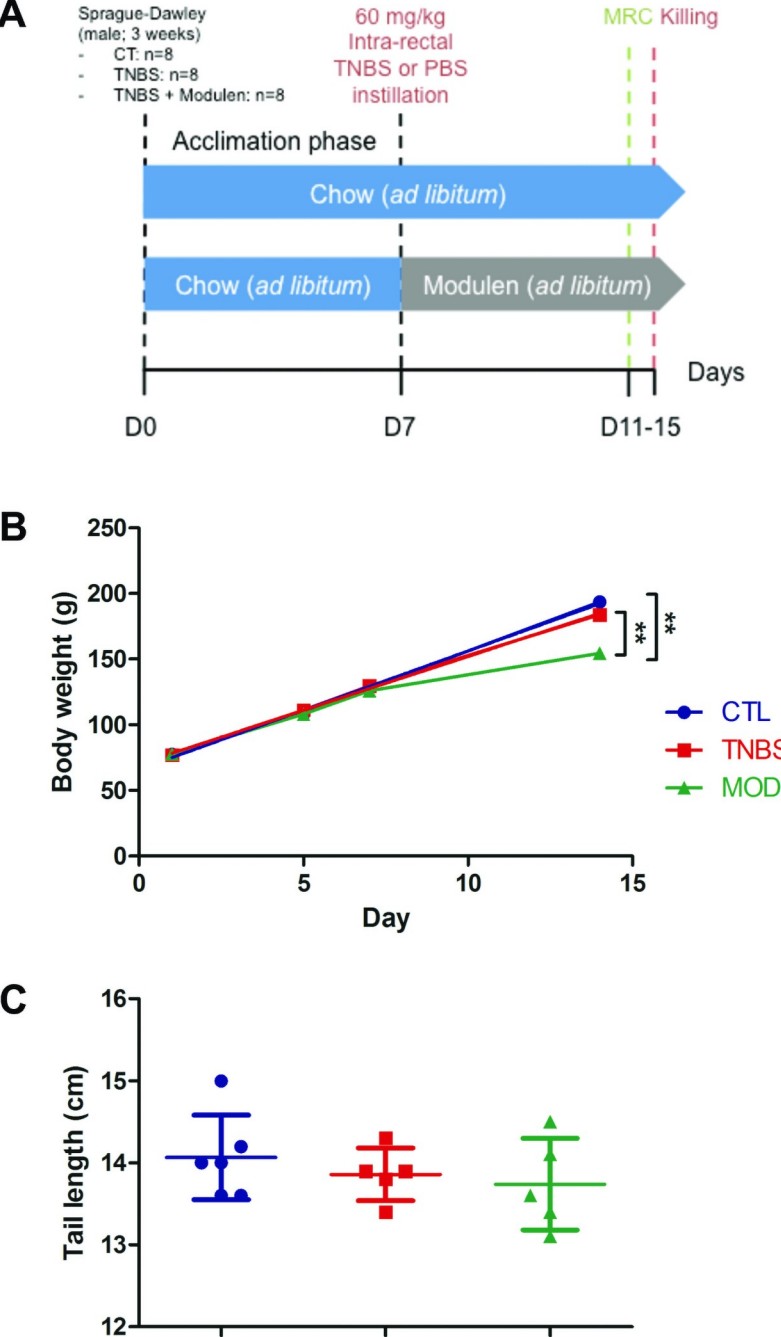

**Fig 1. Experimental design for induction of acute colitis in pre-pubertal rats, and treatment with MODULEN-IBD®.** A. Experimental design. B. Body weight by group, in the days following randomization (** P<0.01 compared to the 2 other groups). C. Tail length at the end of the study, by group (The groups did not differ statistically). CTL: controls, TNBS: 2,4,6-trinitrobenzene sulfonic acid colitis group, MOD: TNBS + MODULEN group.

All sections of this report adhere to the ARRIVE Guidelines for reporting animal research [20]. Sample size was calculated using G*Power Software [21]. Based on our previous study [17], a difference in colon wall thickness on MRC between control and TNBS group was shown with an effect size of 3.7, and mean mortality rate was 30%. We estimated that effect size would be 50% lower (= 1.9) using a dose of TNBS reduced by 50% to take into account higher susceptibility to drugs in young animals. To achieve power = 0.80 and alpha = 0.05 to detect this difference would require 6 animals per group, which we increased to 8 based on a predicted attrition rate of 30%.

## Induction of colitis

After one week of acclimation, colitis was induced by intra-rectal instillation of 2,4,6-trinitro-benzene sulfonic acid (TNBS, Sigma-Aldrich, Saint-Quentin Fallavier, France). After fasting for 24 hours, rats were anesthetized by intra-peritoneal injection of ketamine 8 mg/kg (Pan-pharma) and Chlorpromazine 1 mg/kg (Sanofi-Aventis). A polyurethane cannula was inserted 3 cm into the rectum and TNBS—ethanol injected, at a dose of 60 mg/kg of body weight. TNBS was mixed with a 50% Ethanol vehicle, for a total volume of 250 µL. The rats were maintained in a Trendelenburg position for 5 minutes after the injection to minimize leaks. Rats in the control group received an intra-rectal instillation of 250 µL of phosphate-buffered saline (PBS). During the week following induction of colitis, the rats were observed for signs of pain or significant weight loss.

## Magnetic resonance colonography

MRC was performed within the week following TNBS or PBS administration (between Day 4 and Day 7), using a small animal machine: Bruker BioSpec 47 / 40 USR, 4.7 Tesla (Bruker Biospin, Ettlingen, Germany). Rats were anesthetized by intra-peritoneal injection of thiopental 90 mg/kg (Panpharma). Cardiac rhythm was monitored by surface electrodes. Rats were installed in the cradle in a supine position. There was no injection of antispasmodic or contrast agent. Respiratory movements were corrected using the Intragate™ technique.

Parameters for the T2 rapid acquisition with relaxation enhancement (RARE) sequence were: Repetition time (RT) 5443 ms, Echo time (ET) 34 ms, matrix 320x224, slice 1 mm, Number of excitations (NEX) 3, flip angle 180˚, field of view (FOV) 5.0 x 5.3 cm, acquisition time: 10 minutes, RARE factor 8. Parameters for T2 RARE with fat suppression (FAT-SAT): ET 39 ms, RT 6027 ms, RARE factor 8, NEX 3, acquisition time 8 minutes, flip angle 180˚, slice 1 mm, FOV 5.0 x 5.3 cm, matrix 320x224. T1 sequence with intragate fast low angle shot (FLASH); RT 413 ms, ET 2.8 ms, matrix 256x256, slice 1.1 mm, flip angle 80˚, FOV 4.5 x 4.5 cm; acquisition time: 14 minutes. MRC images were analysed in DICOM, using ParaVision 5.0 software. A senior radiologist (C. S.-C.) who was blind to allocation group interpreted all images.

Image quality taking into account respiratory and bowel wall movements was assessed on a scale of 0 (poor quality) to 3 (excellent quality). Measures were made in the descending colon.

In order to assess inflammation, the criteria used were: maximal colon wall thickness (average of 3 measures), minimal colon wall thickness (average of 3 measures), colon wall thickness at splenic angle (average of 3 measures), colon wall signal intensity in Regions of Interest (ROI) on T2w sequences (average of 2 measures), target sign, colon mucosa irregularities suggesting ulcerations, spontaneous enhancement of colon wall T1w signal [17].

## Killing and samples

On Day 8 after colitis induction, rats were killed by a lethal dose of intraperitoneal thiopental and then decapitated. Tail was measured as a proxy for growth. Colon was sampled, washed

with PBS to remove feces, measured and weighed. Six one-centimeter samples were taken from the colon, starting from the rectum, one of which was fixed in 10% neutral buffered formalin (Sigma-Aldrich) for histopathology, the others stored at– 80˚C until analysis.

## Histopathology

Histological analyses were made by a senior pathologist (M.A.), blinded to allocation group. The formalin-fixed samples were embedded in paraffin, and 5-micrometer sections were colored with hematoxylin/eosin (H&E; Merck, Darmstadt, Germany) for standard histopathology. Samples were studied on 3 levels of cut. Inflammation was scored using a semi-quantitative score previously used by our team: from 0 (no inflammation) to 3 (severe inflammation) [17, 22]. Fibrosis was scored from 0 (no fibrosis) to 3 (severe fibrosis). Images were taken by standard light microscopy using a Leica microscope.

## Colon expression of cyclooxygenase-2 (COX-2) by Western Blot

Frozen colon samples were homogenized in PBS with 0.1% protease inhibitor cocktail (Sigma) and phosphatase inhibitor cocktail (Sigma). Homogenates were centrifuged (12 000g, 15 min, 4˚C) and supernatants were collected and stored at -80˚C. Protein concentration was determined following Bradford's colorimetric method and Western blot for COX-2 was performed as previously [23]. Protein expression was quantified by densitometry with the ImagQuantTL software (GE Healthcare, USA). To check equal loading, the blots were analyzed for glyceraldehyde-3-phosphate dehydrogenase (GAPDH) expression.

## mRNA levels for Interleukin-1 β (IL-1 β), Tumor necrosis factor α (TNF-α) cytokines and inflammatory marker Nitric Oxide Synthase 2 (NOS2)

Quantitative reverse transcription PCR (RT-qPCR) was performed as previously [24]. Briefly, colon samples were frozen in liquid nitrogen and stored at -80˚C before ribonucleic acid (RNA) preparation. Total RNA was isolated using guanidium isothiocyanate method and reverse transcribed into complementary deoxyribonucleic acid (cDNA). PCR was performed with CFX96 Real-Time System (Bio-Rad, Marnes-la-Coquette, France).

## Data analyses

Characteristics of the rats were compared between groups. Since group sizes were < 30, non-parametric methods were used for analyses: qualitative variables were compared using Fisher's exact test, quantitative variables were compared using Wilcoxon Mann-Whitney with exact correction for small samples. Missing data (following premature death of an animal) was excluded from analyses.

Statistical analyses were performed using SAS 9.2 (Cary, NC, USA) and Graph Pad Prism 5 (San Diego, CA, USA). A two-sided p-value < 0.05 was considered significant.

## Compliance with ethical standards

Animal care and experimentation complied with French and European Community regulations (Directive 2010/63/UE). Study protocol was approved by the Institutional Care and Use Committee (Comité d'Ethique Normande en Matière d'Expérimentation Animale, CENO-MEXA). Painful procedures were carried out under deep sedation and all efforts were made to minimize suffering.

## Results

### Validation of the acute colitis model

**Growth and histopathology.** Body weight growth curve during study protocol and tail length at the end of the protocol did not differ between TNBS and Control groups (Fig 1, Panel B, C). Mortality rate was 25% (2/8) in the control group, and 38% (3/8) in the TNBS group.

Histological inflammation score was significantly higher in the TNBS group compared to control group (p = 0.015) (Fig 2, Panel D). TNBS group had a higher fibrosis score but did not reach statistical significance (p = 0.06) (Fig 2, Panel E). Colon weight/length, a marker of inflammation, did not differ between TNBS and Control groups (p = 0.08) (Fig 2, Panel F), although colon weight was significantly increased in TNBS group compared to Controls (2.3 ± 1.1 g vs. 1.4 ± 0.3 g, p = 0.009).

**Magnetic resonance colonography.** Bowel wall thickness measured in descending colon on axial plane, T2 sequence, was significantly increased in the TNBS group compared to controls (Table 1 and Fig 3, Panel A). MRC signs suggestive of inflammation, such as a target sign, wall ulcers, and increased wall signal intensity on T2w images were significantly more prevalent in the TNBS group than in controls (Table 1). There was no significant increase of T1w

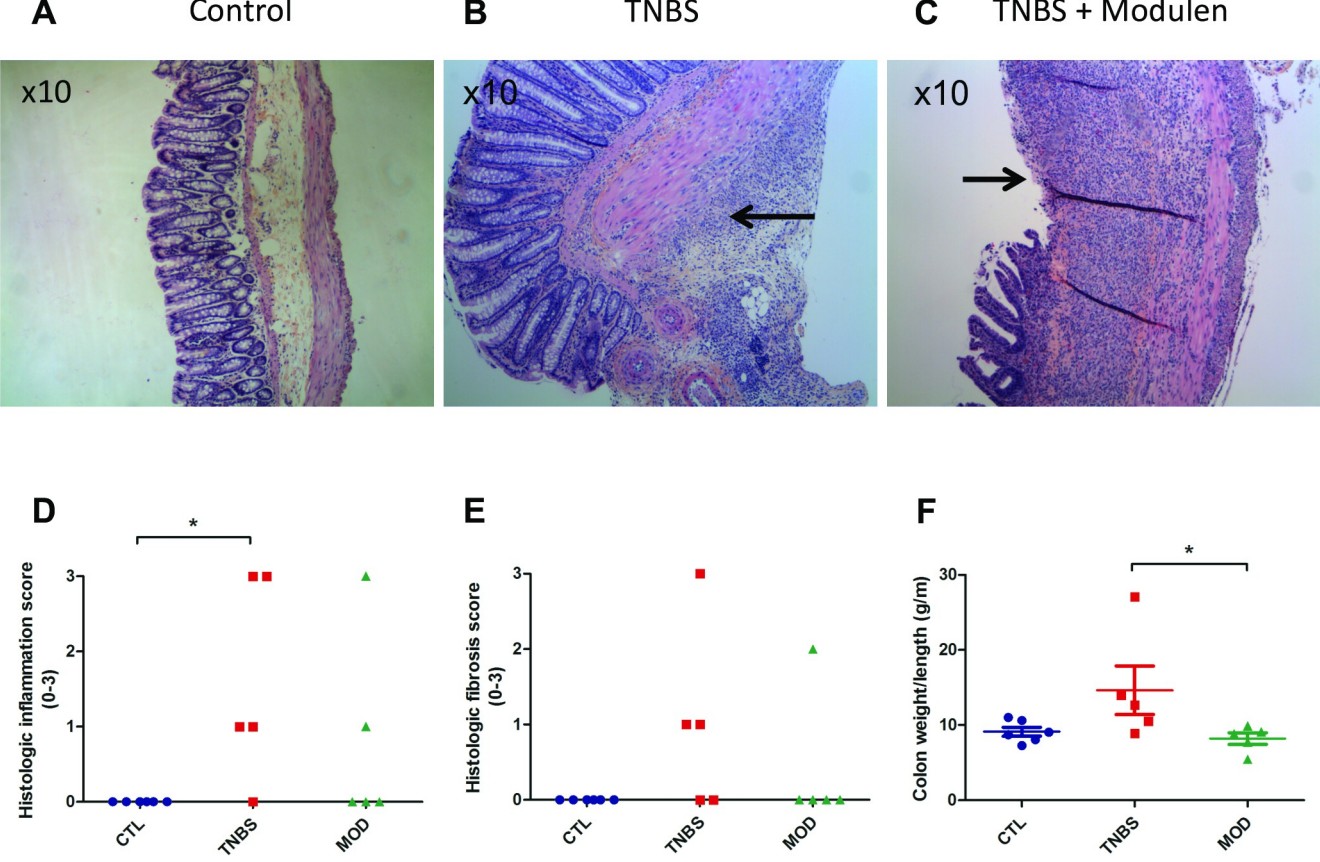

**Fig 2. Validation of the acute colitis model: Histopathology, hematoxylin-eosin stain.** A. H&E histology from the control group showing normal histology. B. Representative histology from the TNBS group. The arrow indicates severe transmural inflammation. C. Representative histopathology from the TNBS + MODULEN. The arrow indicates an ulceration. D. Histology inflammation score by group. E. Histological fibrosis score by group. F. Colon weight/length ratio by group. CTL: controls, TNBS: 2,4,6-trinitrobenzene sulfonic acid colitis group, MOD: TNBS + MODULEN group. * P < 0.05.

**Table 1. Magnetic resonance colonography characteristics—TNBS induced acute colitis model.**

| | CTL (n = 8) | TNBS (n = 5) | MOD (n = 6) | TNBS/ CTL (p) | MOD/ TNBS (p) | MOD/ CTL (p) |
|---|---|---|---|---|---|---|
| **Maximum colon wall thickness (axial T2) (mm)** | 0.36 ±0.04 | 1.03 ±0.36 | 1.26 ±0.48 | 0.002 | NS | <0.001 |
| **Minimal wall thickness (mm)** | 0.36±0.05 | 0.63±0.14 | 0.69±0.19 | 0.002 | NS | <0.001 |
| **Wall thickness at kidney hilum level (mm)** | 0.42 ±0.06 | 0.72 ±0.14 | 0.53 ±0.22 | 0.002 | NS | *0.06* |
| **Increased wall signal intensity on T2w (n, %)** | 0 (0) | 4 (0.80) | 5 (0.83) | 0.007 | NS | 0.003 |
| **Target sign—yes (n, %)** | 0 (0) | 4 (0.80) | 3 (0.50) | 0.007 | NS | *0.055* |
| **Spontaneous T1w hypersignal intensity (n, %)** | 0 (0) | 2 (0.40) | 1 (0.17) | NS | NS | NS |
| **Irregular patterns of mucosal wall (ulcers) (n, %)** | 0 (0) | 5 (1.0) | 6 (1.0) | <0.001 | - | <0.001 |
| **Stenosis (n, %)** | 0 (0) | 3 (0.60) | 6 (1.0) | 0.04 | NS | <0.001 |
| **Mucosal flap (n, %)** | 0 (0) | 0 (0) | 2 (0.33) | - | NS | NS |

wall signal intensity, luminal stenosis or mucosal flap. Image quality was good, except for 1 rat in the TNBS group that had wall artifacts.

**Inflammatory markers and cytokine expression.** The mRNA levels of 3 inflammatory markers genes (IL1β, TNFα and NOS2) were not significantly different between TNBS and control groups (Fig 4). Colon expression of COX-2 was not significantly different between groups (Fig 4).

## Effect of MODULEN-IBD® in an acute TNBS colitis model

**Growth.** Final body weight in the TNBS+MODULEN group was significantly lower than TNBS group (154.6 ± 12.9 g vs. 183.8 ± 14.6, p = 0.004) and control group (mean weight 193.6 ± 11.9 g, p = 0.0007) (Fig 1, Panel B). Tail length did not differ significantly between groups (Fig 1, Panel C).

**Histopathology.** Histological inflammation and fibrosis scores did not differ significantly between TNBS+MODULEN and TNBS group (Fig 2, Panel D). Colon weight/length ratio, a marker of inflammation, was significantly lower in the TNBS+MODULEN group than in the TNBS group (p = 0.03) (Fig 2, Panel F).

**Magnetic resonance colonography.** There was no significant difference in MRC characteristics between TNBS and TNBS+MODULEN groups (Table 1). Rats from the TNBS+MODULEN group differed from controls on the following characteristics: increased wall thickness, increased wall intensity on T2w images, presence of ulcers (100% of rats) and luminal stenosis (100% of rats). A mucosal flap was observed in 33% of rats from the TNBS+MODULEN group, but not in other groups. Image quality was good, except for 1 rat that had wall artifacts and 1 that had respiratory artifacts.

**Inflammatory marker and cytokine expression.** The colon mRNA expression of IL-1β, TNF-α and NOS2 was not significantly different between MODULEN and TNBS or control groups (Fig 4). Colon expression of COX-2 was not significantly different between groups.

## Discussion

In this study we have shown that a single intra-rectal instillation of TNBS was able to induce acute colitis in pre-pubertal rats, and that typical features of acute colitis could be observed on MRC and confirmed by histopathology.

The TNBS-hapten model of colitis first described in 1989 by Morris *et al.* [12] has been shown to mimic IBD by eliciting a Th1 T-cell based response. Although this model has been frequently used for studies in adult rats, few protocols exist in pre-pubertal rats. Hence our

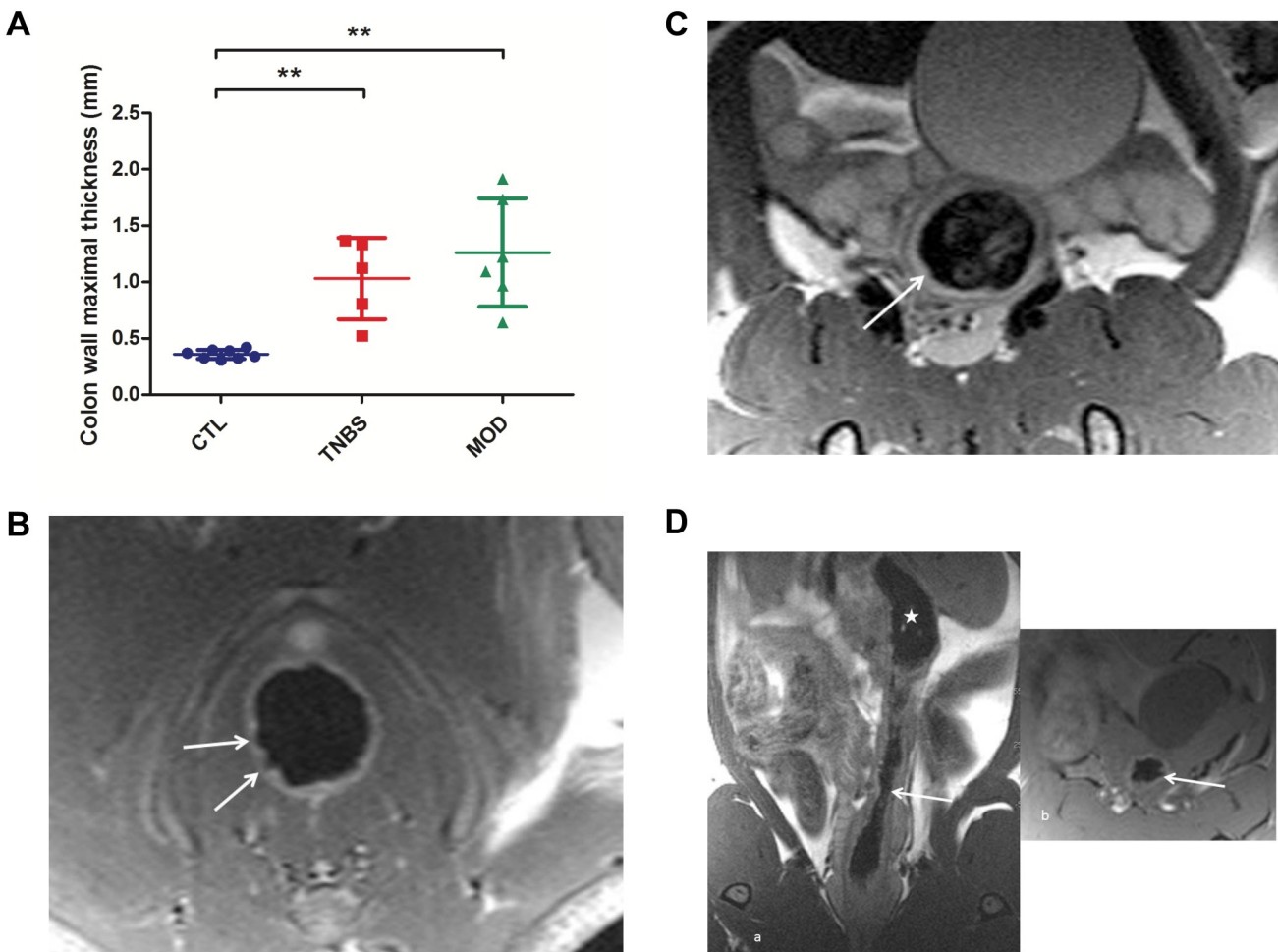

**Fig 3. Validation of acute colitis model: Magnetic resonance colonography.** A. Colon wall thickening: maximum thickness, by group (** P < 0.01). B. MRC representative image from the TNBS group. The arrow indicates an irregular colon wall suggestive of an ulcer. C. MRC representative image from the TNBS group: the arrow indicates increased wall signal intensity on T2-weighted sequence. D. MRC representative image from the TNBS group: the arrow indicates narrowing of the colon lumen suggestive of a stenosis and the star, a pre-stenotic dilation. MRC: magnetic resonance colonography. CTL: controls, TNBS: 2,4,6-trinitrobenzene sulfonic acid colitis group, MOD: TNBS + MODULEN group.

first objective was to adapt this model of acute experimental colitis to pre-pubertal rats. There is no standardized protocol for inducing TNBS colitis in rats, and various doses of TNBS have been used, putting into balance the severity of colitis induced with the mortality rate [25]. Low doses of 50 mg/kg cause mild colitis (Wallace score of 2), whereas high doses of up to 150 mg/kg cause severe colitis (minimal Wallace score of 5) but are associated with mortality rates of 30% caused by excessive inflammation. The only pre-pubertal rat models reported to date used a fixed dose of 8 mg of TNBS (between 80 and 133 mg/kg depending on animal's weight) [26, 27]. In our experimental protocol we chose an individually weight-adapted dose of 60 mg/kg, to take into account the increased susceptibility caused by young age [15]. This induced mild to severe histological colitis in 4/5 rats of the TNBS group. Despite the low dose chosen, our team's expertise using this model with older animals and precautious manipulations, there was significant mortality (between 2 and 3 rats per group), which underlines the challenge of developing a reproducible model of acute bowel inflammation in pre-pubertal rats with the least invasive procedures possible.

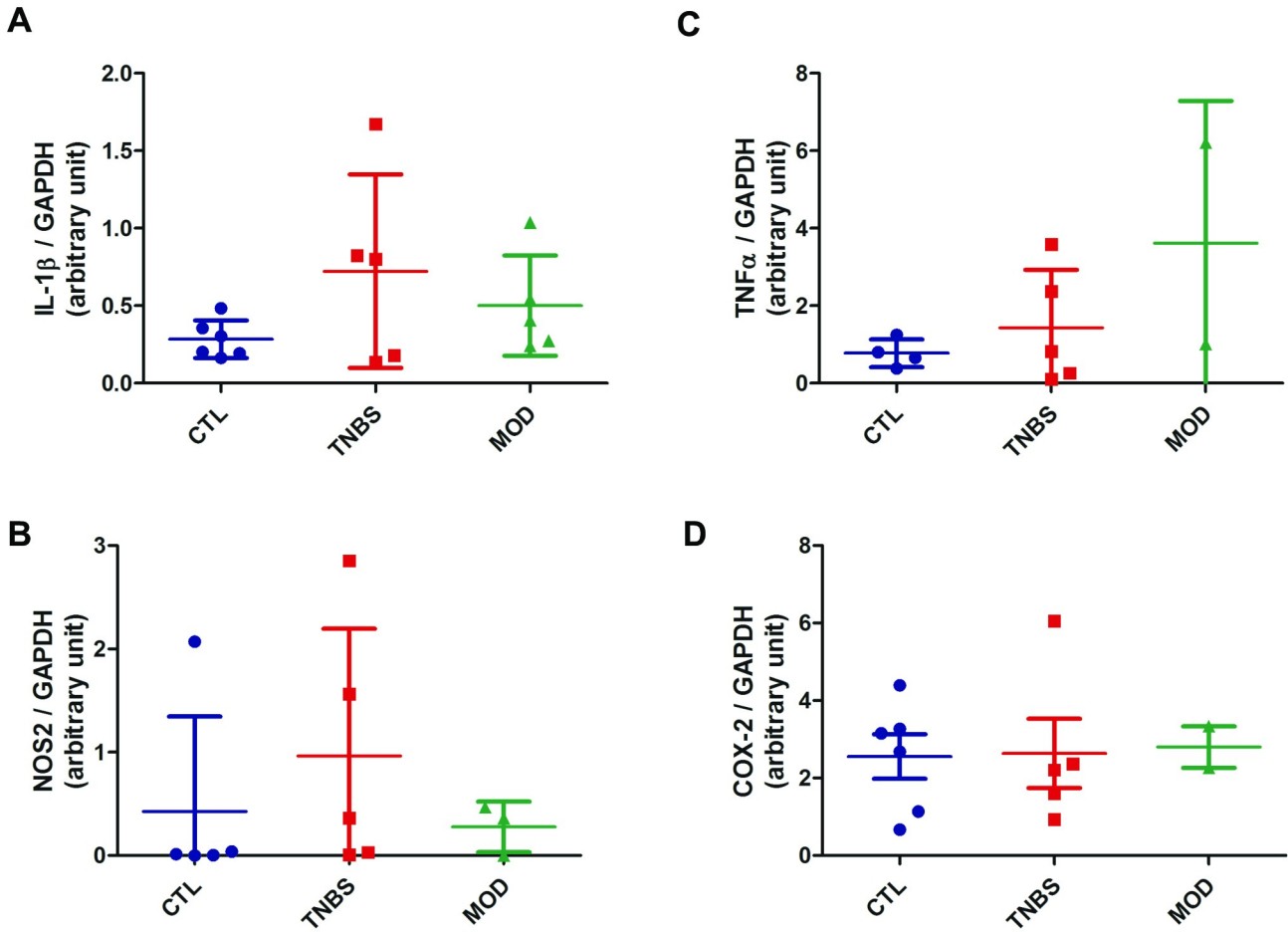

**Fig 4. Expression of inflammatory markers in colonic tissue homogenates, TNBS-induced acute colitis model in pre-pubertal rats.** Gene expression measured by RT-PCR: IL-1β (Panel A), NOS2 (Panel B) and TNF-α (Panel C); Expression of COX-2 measured by Western Blot (Panel D). None of the groups were statistically different. CTL: controls, TNBS: 2,4,6-trinitrobenzene sulfonic acid colitis group, MOD: TNBS + MODULEN group.

Magnetic resonance imaging is an important technique to follow progression of inflammation and tissue damage in IBD. Our team showed the feasibility of MRC to assess acute TNBS colitis in adult rats [17]. Consistently with our previous study, we obtained good quality images, without requiring anti spasmodic or contrast agents, and observed typical signs of colon inflammation in TNBS treated rats, but not in controls: ulcers, target sign and bowel wall thickening.

We were not able to replicate our previous findings of increased colon production of IL-1β and COX-2 following TNBS administration. A possible explanation for this discrepancy was that we chose to wait one week after induction of colitis instead of 2 days, hoping to observe the maximum severity of lesions. However, this delay might have allowed the pro-inflammatory markers to normalize, contemporary to recovery process, although histological lesions might take longer to.

We were surprised by a signal of fibrosis observed on histopathology, since rats in the TNBS group had a higher although not significant histological fibrosis score than controls. Intestinal fibrosis is a dynamic process, which is thought to arise as a consequence of chronic uncontrolled inflammation. Fibrosis in TNBS models has been observed after 4 to 8 doses of TNBS [27, 28].

Presence of fibrosis as soon as 1 week after the first dose of TNBS would suggest that tissue repair pathways might be activated very early in inflammation or might be an independent phenomenon. Brenna *et al.* showed temporal changes in gene expression in acute TNBS colitis between 0 and 12 days after induction of colitis, with a decrease in regulation of metabolism and an increase of the regulation of tissue remodeling genes [29]. They conclude that mucosal cells might be exposed to pro-fibrotic signaling cascades during inflammatory response.

The anti-inflammatory effect of MODULEN-IBD® has been shown in children based on clinical, biological and endoscopic endpoints [30–33]. In our pre-pubertal rat model, we showed that the histological inflammation scores of TNBS rats treated with MODULEN-IBD® did not differ from controls or from TNBS group. In addition, the reduced colon weight/length ratio in the TNBS+MODULEN group is in favor of an anti-inflammatory effect. It must be noted that reduced weight/length ratio cannot be attributed to growth failure, since colon and tail lengths were not different between groups. We did not observe different final weights between TNBS and control groups, based on the body weight at the end of the protocol, 8 days after induction of colitis. Weight loss is one of the clinical features of the TNBS colitis model. However, weight loss reaches a peak 3 days after induction of colitis but can be recovered by Day 7 [29], which could explain the absence of weight difference between TNBS and control groups. However, final weight was lower in the TNBS + MODULEN group than in the two others, indicating that the intake of MODULEN-IBD® might have been insufficient to meet the energy requirements. The macronutrient composition of MODULEN-IBD® has several differences with a standard growing rodent diet. Energy content of MODULEN-IBD® is 5000 kcal / kg of powder, of which 42% is brought by lipids, 44% by carbohydrates and 14% by protein. In contrast, the standard A03 "rodent diet for growing and breeding" (SAFE) nutritional composition is: 13.5% lipids, 61.3% carbohydrates, 25.2% protein with a total of 3395 kcal / kg of powder form. Given as a sole source of nutrition, and ad libitum instead of by oral gavage, the effect might also have been hampered by low palatability. In order to control for feeding difficulties due to the powdered form of the feed, we fed the control and TNBS groups the standard rodent diet in a powdered form and not as pellets.

We did not observe a difference in levels of mRNA for inflammatory cytokines (IL-1β and TNFα) or NOS 2 and colon COX-2 expression between the 3 groups, although this could be due to a lack of power due to small size of study groups. These inflammatory markers are part of the NF-κB signaling pathway [34]. In our previous studies with older rats, we found that acute TNBS colitis was associated with colon NF-κB activation with upregulated downstream molecules [35]. This was in accordance with findings from pre-clinical studies using chemical-induced colitis [11, 36]. As in the present study, we previously did not observe a significant difference in colon COX-2 expression in rats with TNBS-induced colitis 7 days after the TNBS injection while colon COX-2 expression was upregulated in rats with TNBS-induced colitis after 2 days [17]. The timing of the present study (7 days after) may have contributed to this discrepancy by a partial recovery.

Treatment with MODULEN-IBD® did not reverse the intestinal damage seen on MRC. This might have been affected by the timing of MRI. Rimola *et al.* showed in adult Crohn's disease patients persistent damage on MR-enterography (MRE) despite endoscopic remission after one year of anti-TNF or stem cell transplantation treatment [37]. The MRE abnormalities that persisted were: mural hyperenhancement, mural thickness and strictures. One might argue that the persistence of strictures reflects fibrotic sequelae of inflammation. In animal models, the minimal delay in which improvement and resolution of lesions on MRC can be expected remains to be determined, using longitudinal repeated MRC measures. Future studies could also incorporate analysis of gut microbiota, in particular since MODULEN-IBD® has been shown to modify gut microbiota composition in children [33].

In conclusion, we have shown that a single intra-rectal installation of TNBS in prepubertal rats causes colonic inflammation that can be observed on histopathology and non-contrast MRC. Treatment with MODULEN-IBD® showed a mild anti-inflammatory effect on histopathology, but not on MRC. Optimal timing of MRC remains to be determined before implementing MRC as routine non-invasive assessment tool of therapeutic interventions efficacy in TNBS colitis.

## Acknowledgments

The authors would like to thank Elodie Colasse and Amelyne David for their assistance in data acquisition.

## Author Contributions

**Conceptualization:** Claire Dupont-Lucas, Rachel Marion-Letellier, Guillaume Savoye.

**Data curation:** Claire Dupont-Lucas, Rachel Marion-Letellier.

**Formal analysis:** Claire Dupont-Lucas, Mathilde Pala.

**Investigation:** Mathilde Pala, Charlène Guerin, Christine Bôle-Feysot, Emmeline Salameh, Alexis Goichon, Lionel Nicol, Moutaz Aziz, Céline Savoye-Collet.

**Methodology:** Claire Dupont-Lucas, Rachel Marion-Letellier, Guillaume Savoye.

**Project administration:** Charlène Guerin.

**Supervision:** Rachel Marion-Letellier, Guillaume Savoye.

**Writing – original draft:** Claire Dupont-Lucas.

**Writing – review & editing:** Claire Dupont-Lucas, Rachel Marion-Letellier, Mathilde Pala, Charlène Guerin, Christine Bôle-Feysot, Emmeline Salameh, Alexis Goichon, Lionel Nicol, Moutaz Aziz, Céline Savoye-Collet, Guillaume Savoye.

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
