## [Decision Letter · Decision Letter 0]

5 Aug 2021

PONE-D-21-22589

Magnetic Resonance Colonography assessment of Acute Trinitrobenzene Sulfonic Acid Colitis in pre-pubertal rats

PLOS ONE

Dear Dr. Dupont-Lucas, 

Thank you for submitting your manuscript to PLOS ONE. After careful consideration, we feel that it has merit but does not fully meet PLOS ONE’s publication criteria as it currently stands. Therefore, we invite you to submit a revised version of the manuscript that addresses the points raised during the review process.

We look forward to receiving your revised manuscript.

Kind regards,

Rosanna Di Paola, MD

Academic Editor

PLOS ONE

Journal Requirements:

Reviewers' comments:

Reviewer's Responses to Questions

Comments to the Author

1. Is the manuscript technically sound, and do the data support the conclusions?

Reviewer #1: Partly

2. Has the statistical analysis been performed appropriately and rigorously?

Reviewer #1: Yes

3. Have the authors made all data underlying the findings in their manuscript fully available?

Reviewer #1: Yes

4. Is the manuscript presented in an intelligible fashion and written in standard English?

Reviewer #1: Yes

5. Review Comments to the Author

Reviewer #1: This study investigated the Magnetic Resonance Colonography assessment of Acute Trinitrobenzene Sulfonic

Acid Colitis in pre-pubertal rats. The rational behind the experiment was clear and

straight forward. The manuscript is almost well written.

While many different sources are used to set up the study in the introduction, little previous evidence is

stated. The introduction is thus short and poorly sets up the rationale for the study. More attention to how

this study fits into previous work in colitis and inflammation should be added to improve this section.

Please refer to doi: 10.3390/nu12030834, 10.3109/08830180009048389

There are some minor grammar issues that should be fixed in order to aid the accessibility of the results to

the reader.

The discussion does a good job at explaining the importance of the results in the context of the inflammatory

pathways involved. However, incorporation of previous results from other related studies is lacking.

Incorporating comparisons with other studies would increase the strength of the paper. Please describe

better the role of the inflammation pathway. Please refer to: doi:10.1016/j.phrs.2019.01.041, 10.1007/s10787-016-0263-6.

6. PLOS authors have the option to publish the peer review history of their article (what does this mean?). If published, this will include your full peer review and any attached files.

Do you want your identity to be public for this peer review? For information about this choice, including consent withdrawal, please see our Privacy Policy.

Reviewer #1: No

---

## [Author Response · Author response to Decision Letter 0]

28 Aug 2021

Here are the point-by-point replies to the reviewer’s comments:

Reviewer #1: This study investigated the Magnetic Resonance Colonography assessment of Acute Trinitrobenzene Sulfonic Acid Colitis in pre-pubertal rats. The rational behind the experiment was clear andstraight forward. The manuscript is almost well written.

=> Thanks for your comments

While many different sources are used to set up the study in the introduction, little previous evidence is

stated. The introduction is thus short and poorly sets up the rationale for the study. More attention to how

this study fits into previous work in colitis and inflammation should be added to improve this section.

Please refer to doi: 10.3390/nu12030834 [Siracusa Nutrients 2020], 10.3109/08830180009048389 [Neurath Int Rev Immunol 2000]

=> we thank the reviewer for these references and have added them to the introduction:

- Lines 58-60: Added reference: Siracusa Nutrients 2020 [11]

“Animal models of intestinal inflammation mimicking Inflammatory Bowel Disease (IBD) are widely used to better understand the cellular and molecular pathways of inflammation and fibrosis, and target these pathways to develop new drugs [8-11].”

- Lines 61-63: Added reference: Neurath Int Rev Immunol 2000 [13] 

“Intra-rectal instillation of TNBS causes an IL-12-driven Th1 T-cell mediated immune response in the colonic mucosa [13].”

There are some minor grammar issues that should be fixed in order to aid the accessibility of the results to

the reader.

=> English language was checked by a native-speaker and we made the necessary grammar corrections

The discussion does a good job at explaining the importance of the results in the context of the inflammatory

pathways involved. However, incorporation of previous results from other related studies is lacking.

Incorporating comparisons with other studies would increase the strength of the paper. Please describe

better the role of the inflammation pathway. Please refer to: doi:10.1016/j.phrs.2019.01.041 [Di Paola Pharmacol Res 2019], 10.1007/s10787-016-0263-6.

=> these references have been added to the introduction and discussion: 

- Lines 366-367 : Added reference : Di Paola Pharmacol Res 2019 [36]

“This was in accordance with findings from pre-clinical studies using chemical-induced colitis [11,36].”

- Lines 70-72: Added reference Rashidian Inflammopharmaco 2016 [14]

“The model has been used to test anti-inflammatory properties of dietary compounds, and drugs on colonic inflammation through inhibition of the NF-�B pathway [11, 14].” 

We also added a paragraph on the inflammation pathway with comparisons with studies from others and us

Lines 361-371:

“We did not observe a difference in levels of mRNA for inflammatory cytokines (IL-1� and TNF�) or NOS 2 and colon COX-2 expression between the 3 groups, although this could be due to a lack of power due to small size of study groups. These inflammatory markers are part of the NF-�B signaling pathway [34]. In our previous studies with older rats, we found that acute TNBS colitis was associated with colon NF-�B activation with upregulated downstream molecules [35]. This was in accordance with findings from pre-clinical studies using chemical-induced colitis [11, 36]. As in the present study, we previously did not observe a significant difference in colon COX-2 expression in rats with TNBS-induced colitis 7 days after the TNBS injection while colon COX-2 expression was upregulated in rats with TNBS-induced colitis after 2 days [17]. The timing of the present study (7 days after) may have contributed to this discrepancy by a partial recovery. “

---

## [Editor Report · Decision Letter 1]

14 Oct 2021

Magnetic Resonance Colonography assessment of Acute Trinitrobenzene Sulfonic Acid Colitis in pre-pubertal rats

PONE-D-21-22589R1

Dear Dr. 

We’re pleased to inform you that your manuscript has been judged scientifically suitable for publication and will be formally accepted for publication once it meets all outstanding technical requirements.

Kind regards,

Rosanna Di Paola, MD

Academic Editor

PLOS ONE
---

## [Editor Report · Acceptance letter]

18 Oct 2021

PONE-D-21-22589R1 

Magnetic Resonance Colonography assessment of Acute Trinitrobenzene Sulfonic Acid Colitis in pre-pubertal rats 

Dear Dr. Dupont-Lucas:

I'm pleased to inform you that your manuscript has been deemed suitable for publication in PLOS ONE. Congratulations! Your manuscript is now with our production department. 

Kind regards, 

on behalf of

Dr. Rosanna Di Paola 

Academic Editor

PLOS ONE